# CemR atypical response regulator impacts energy conversion in *Campylobacteria*

Mateusz Noszka,[1] Agnieszka Strzałka,[2] Jakub Muraszko,[1] Dirk Hofreuter,[3] Miriam Abele,[4] Christina Ludwig,[4] Kerstin Stingl,[5] Anna Zawilak-Pawlik[1]

**ABSTRACT** *Campylobacter jejuni* and *Arcobacter butzleri* are microaerobic food-borne human gastrointestinal pathogens that mainly cause diarrheal disease. These related species of the *Campylobacteria* class face variable atmospheric environments during infection and transmission, ranging from nearly anaerobic to aerobic conditions. Consequently, their lifestyles require that both pathogens need to adjust their metabolism and respiration to the changing oxygen concentrations of the colonization sites. Our transcriptomic and proteomic studies revealed that *C. jejuni* and *A. butzleri,* lacking a *Campylobacteria*-specific regulatory protein, *C. jejuni* Cj1608, or a homolog, *A. butzleri* Abu0127, are unable to reprogram tricarboxylic acid cycle or respiration pathways, respectively, to produce ATP efficiently and, in consequence, adjust growth to changing oxygen supply. We propose that these *Campylobacteria* energy and metabolism regulators (CemRs) are long-sought transcription factors controlling the metabolic shift related to oxygen availability, essential for these bacteria's survival and adaptation to the niches they inhabit. Besides their significant universal role in *Campylobacteria*, CemRs, as pleiotropic regulators, control the transcription of many genes, often specific to the species, under microaerophilic conditions and in response to oxidative stress.

**IMPORTANCE** *C. jejuni* and *A. butzleri* are closely related pathogens that infect the human gastrointestinal tract. In order to infect humans successfully, they need to change their metabolism as nutrient and respiratory conditions change. A regulator called CemR has been identified, which helps them adapt their metabolism to changing conditions, particularly oxygen availability in the gastrointestinal tract so that they can produce enough energy for survival and spread. Without CemR, these bacteria, as well as a related species, *Helicobacter pylori*, produce less energy, grow more slowly, or, in the case of *C. jejuni*, do not grow at all. Furthermore, CemR is a global regulator that controls the synthesis of many genes in each species, potentially allowing them to adapt to their ecological niches as well as establish infection. Therefore, the identification of CemR opens new possibilities for studying the pathogenicity of *C. jejuni* and *A. butzleri*.

**KEYWORDS** *Campylobacter jejuni*, *Arcobacter butzleri*, carbon metabolism, respiration, transcription factors, proteomics, transcriptomics, oxidative stress, *Helicobacter pylori*, signal transduction

Address correspondence to Anna Zawilak-Pawlik, anna.pawlik@hirszfeld.pl.

The authors declare no conflict of interest.

See the funding table on p. 22.

*C*ampylobacter jejuni and *Arcobacter butzleri* are Gram-negative, microaerobic bacteria that belong to the *Campylobacteria* class (1). *C. jejuni* is a commensal bacterium of the gastrointestinal tracts of wildlife and domestic animals. However, in humans, *C. jejuni* is the leading cause of food-borne bacterial diarrheal disease (2). Also, it causes autoimmune neurological diseases such as Guillian-Barré and Miller-Fisher syndromes (3). *A. butzleri* is found in many ecological niches, such as environmental water and animals. Still, in humans, *A. butzleri* causes diarrhea, enteritis, and bacteremia (2). Due to the wide distribution of the

10.1128/msystems.00784-24 **1**

two species and relatively high resistance to environmental conditions (4) and antibiotics (5, 6), both species pose a severe threat to human health.

To adapt to different ecological niches and decide whether conditions are optimal for reproduction and transmission, *C. jejuni* and *A. butzleri* encode numerous regulatory proteins. According to the MiST database (7), *C. jejuni* NCTC 11168 [1.64 Mbp; 1,643 coding sequences (8)] and *A. butzleri* RM4018 [2.34 Mbp; 2,259 coding sequences (9)] encode 56 and 218 signal transduction proteins, respectively. While *A. butzleri* regulatory systems have been hardly studied (10), the *C. jejuni* signal transduction systems have been partially deciphered (11–15). Particular interest was directed to studying processes related to the *C. jejuni* colonization and focused on factors controlling the expression of genes involved in such processes as host adaptation and niche detection (BumSR and DccSR), oxidative stress (e.g., CosR and PerR), metabolism, and respiration [e.g., BumSR, RacSR, CosR, CsrA, LysR (Cj1000), and CprSR] (11, 13–15). Notably, aerobic or microaerophilic respiration, in which oxygen is used as a terminal electron acceptor, is far more efficient in supplying energy than anaerobic respiration or fermentation (16). On the other hand, oxygen-dependent respiration is a source of reactive oxygen species (ROS) (17). *C. jejuni* requires oxygen for growth but, unlike facultative aerobic *A. butzleri*, is sensitive to aerobic conditions faced during transmission. In addition, *C. jejuni* and *A. butzleri* may encounter conditions deficient in oxygen in the intestine or during intracellular persistence (18). Thus, *C. jejuni* and *A. butzleri* must adjust their metabolism and respiration as oxygen concentrations change at colonization sites to get sufficient energy while maintaining redox homeostasis. However, no orthologs of energy or redox metabolism regulators such as gammaproteobacterial ArcA, FNR, or NarP are found in *C. jejuni* (14, 19–21).

The roles of many *C. jejuni* and most *A. butzleri* regulators have still not been explored. One such regulator is the *C. jejuni* NCTC 11168 Cj1608 orphan response regulator, which is homologous to uncharacterized *A. butzleri* RM4018 Abu0127. These regulators are homologous to *Helicobacter pylori* 26695 HP1021 and are conserved across most *Campylobacteria* class species, constituting most of the *Campylobacterota* phylum.

The *H. pylori* HP1021 response regulator is one of 28 signal transduction proteins encoded by *H. pylori* 26695 (7, 22). HP1021 interacts with the origin of chromosome replication region (*oriC*) *in vitro*, probably participating in the control of the initiation of *H. pylori* chromosome replication (23). HP1021 acts as a redox switch protein, i.e., senses redox imbalance and transmits the signal and triggers the cells' response (24). The HP1021 regulon, initially determined in *H. pylori* 26695 by microarray analyses (25), has been updated in *H. pylori* N6 using a multi-omics approach (26). HP1021 influences the transcription of almost 30% of all *H. pylori* N6 genes of different cellular categories; the transcription of most of these genes is related to response to oxidative stress. HP1021 directly controls typical ROS response pathways and less canonical ones, such as central carbohydrate metabolism. The level of ATP and the growth rate of the knock-out *H. pylori* ΔHP1021 are lower than in the wild-type strain, which is possibly due to reduced transcription of many tricarboxylic acid cycle (TCA) genes and/or increased ATP consumption in catabolic processes in ΔHP1021 compared to the wild-type strain. Thus, HP1021, among many cellular processes, probably controls *H. pylori* metabolic fluxes to maintain the balance between anabolic and catabolic reactions, possibly for efficient oxidative stress response (26).

In this work, we focused on the two hardly characterized regulatory proteins of the microaerobic *C. jejuni* NCTC 11168 and the facultatively aerobic *A. butzleri* RM4018, Cj1608, and Abu0127, respectively. To get insight into the function of these proteins, we constructed mutants lacking these regulators and looked at transcriptomic and proteomic changes under optimal growth and oxidative stress conditions. Our results indicate that Cj1608 and Abu0127, which we named <u>C</u>ampylobacteria <u>e</u>nergy and <u>m</u>etabolism <u>r</u>egulators (CemRs), support energy conservation in bacterial cells by controlling metabolic and respiration pathways in response to oxygen availability.

## RESULTS

### Influence of the Cj1608 regulator on *C. jejuni* gene and protein expression

Cj1608 has been hardly characterized thus far. It is only known that it interacts *in vitro* with the *C. jejuni oriC* region and the promoter of the *lctP* lactate transporter, probably participating in the control of the initiation of the *C. jejuni* chromosome replication and lactate metabolism (27, 28). Therefore, to elucidate the role of the Cj1608 regulator in controlling *C. jejuni* gene expression and oxidative stress response, we performed transcriptome analysis [RNA sequencing (RNA-seq)] of the *C. jejuni* NCTC 11168 wild-type (CJ) and deletion mutant (ΔCj1608) strains under microaerobic growth (CJ, ΔCj1608) and during paraquat-induced oxidative stress (CJ$_S$, ΔCj1608$_S$) (Fig. S1A).

A comparison of *C. jejuni* ΔCj1608 and CJ transcriptomes revealed 380 differentially transcribed genes (Fig. 1A and B; Fig. S2A; S1 Data). The paraquat stress affected the transcription of genes in wild type and ΔCj1608 strains (232 and 123 genes, respectively) (Fig. 1B; Fig. S2B and C). The transcription of 44 genes was similarly induced or repressed in CJ$_S$ and ΔCj1608$_S$ cells (Fig. S2D, red dots). Thus, these genes responded to oxidative stress, but other or additional factors than Cj1608 controlled them. Using ClusterProfiler (29), we performed a Kyoto Encyclopedia of Genes and Genomes (KEGG) enrichment analysis to identify processes affected most by the lack of Cj1608 or under stress. In the ΔCj1608 strain, three KEGG groups were activated (nitrogen metabolism, ribosomes, and aminoacyl-tRNA-biosynthesis). One KEGG group was suppressed, namely, the TCA cycle (Fig. 1C). In the CJ strain under stress conditions, six KEGG groups were suppressed (e.g., oxidative phosphorylation), while four KEGG groups were activated (e.g., carbon metabolism).

Since post-transcriptional regulation is common in bacteria, including species closely related to *C. jejuni* (30, 31), a proteomics approach [liquid chromatography coupled with tandem mass spectrometry (LC-MS/MS)] was applied to detect proteins whose levels changed between analyzed strains or conditions. A total of 1,170 proteins were detected (S1 Data); however, 173 proteins encoded by genes whose transcription changed in ΔCj1608 were not detected by MS (S1 Data). The comparison between proteomes of ΔCj1608 and CJ strains under microaerobic conditions revealed different levels of 156 proteins (Fig. S4A; red, yellow, and green dots). The transcriptomic and proteomic data correlated strongly (Fig. 1D, red dots; Fig. S5A). However, the differences between transcription and translation patterns were observed for many genes, suggesting a post-transcriptional regulation of *C. jejuni* gene expression (Fig. 1D; blue, yellow, and green dots). Under oxidative stress, the expression of only four genes changed, namely, the expression of ferrochelatase HemH (Cj0503c) was downregulated in the CJ$_S$ cells; at the same time, the levels of three proteins were upregulated [the catalase KatA (Cj1385), the atypical hemin-binding protein Cj1386, mediating the heme trafficking to KatA (32), and the periplasmic protein Cj0200c] (Fig. S4B and C). The small number of proteins whose expression was altered under oxidative stress confirmed the known phenomenon of ceasing protein synthesis by bacteria under oxidative stress (26, 33, 34). The ClusterProfiler KEGG enrichment analysis revealed that genes of two KEGG groups, nitrogen metabolism and ribosome, were activated in the ΔCj1608 strain in comparison to CJ, while the genes of three KEGG groups were suppressed (TCA cycle, ABC transporters, and carbon metabolism) (Fig. 1E). It should be noted that the nitrogen metabolism group includes proteins such as GlnA and GltBD, which are involved in glutamate metabolism and TCA cycle (35), whose levels increased more than 10-fold, and Nap and Nrf complexes, constituting an electron transport chain (ETC) system (see below).

To conclude, the transcriptomic and proteomic data revealed that Cj1608 impacts the transcription of 585 genes and the expression of 156 proteins in *C. jejuni* NCTC 11168, with the TCA and nitrogen metabolism KEGG groups most affected.

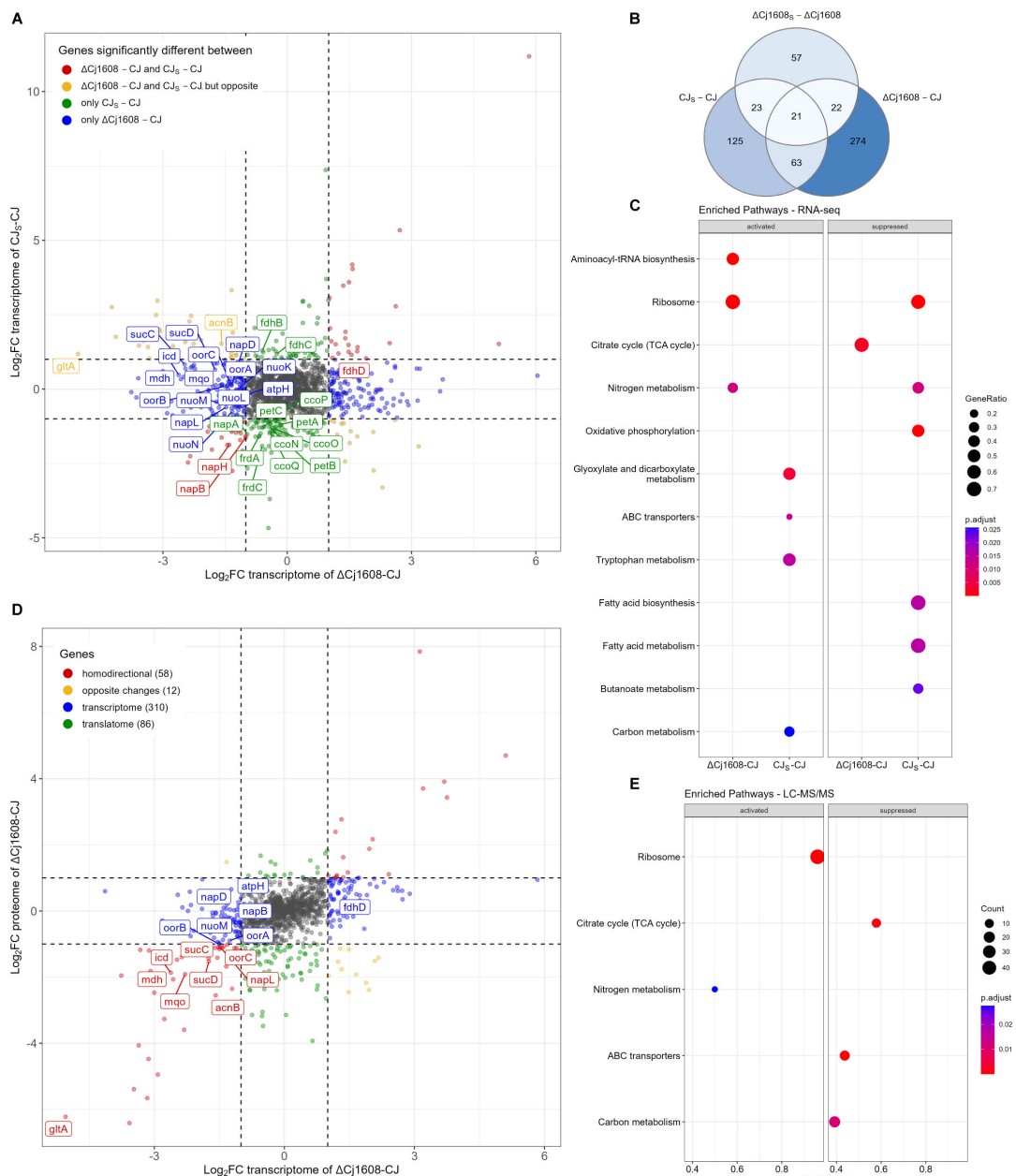

**FIG 1** RNA-seq and LC-MS/MS analyses of *C. jejuni* gene expression controlled by Cj1608. (A) The comparison of gene transcription in the CJ, CJ$_S$, and Cj1608 knock-out mutant (ΔCj1608) cells revealed by RNA-seq. Genes whose transcription significantly changed (|log$_2$FC| ≥ 1, FDR < 0.05) are depicted by colored dots (see the legend in the figure). The genes named on the graph correspond to the citric acid cycle and electron transport chain. (B) The Venn diagram presents the number of differentially transcribed genes in the analyzed strains and conditions. (C) ClusterProfiler protein enrichment plot showing activation or suppression of KEGG groups in the analyzed strains. (D) The correlation between gene expression in ΔCj1608 and CJ cells at the transcription and proteome levels. Red dots represent homodirectional genes up-regulated or down-regulated at the transcriptome and proteome levels. Yellow dots represent opposite changes. Blue dots represent genes that changed only at the transcription level, while green dots represent genes that changed only at the proteome level. Numbers of differentially expressed genes in the indicated strains and conditions are given in parentheses. The proteins named on the graph correspond to the citric acid cycle and electron transport chain. (E) ClusterProfiler protein enrichment plot showing activation or suppression of KEGG groups in the analyzed strains. (A and D) Values outside the black dashed lines indicate a change in the expression of |log$_2$FC| ≥ 1. Gray dots correspond to genes whose transcription was not changed (|log$_2$FC| < 1). CJ, *C. jejuni* wild type; CJ$_S$, stressed wild type; FC, fold change; FDR, false discovery rate; KEGG, Kyoto Encyclopedia of Genes and Genomes; LC-MS/MS, liquid chromatography coupled with tandem mass spectrometry.

## Influence of Abu0127 on *A. butzleri* gene and protein expression

To elucidate the role of the Abu0127 regulator in controlling gene expression and oxidative stress response of *A. butzleri*, we performed RNA-seq and LC-MS/MS analyses similar to those for *C. jejuni*. We analyzed gene expression in *A. butzleri* RM4018 wild-type (AB) and deletion mutant (ΔAbu0127) strains; paraquat was used to induce oxidative stress (AB$_S$, ΔAbu0127$_S$) [Fig. S1B (36)].

Comparison of *A. butzleri* ΔAbu127 and AB transcriptomes revealed 779 differentially transcribed genes (Fig. 2A and B; Fig. S3A; S2 Data). The paraquat stress affected the transcription of genes in the wild type and, to a much lesser extent, in the ΔCj1608 strain (290 and 14 genes, respectively) (Fig. 2B; Fig. S3B and C). Of these 14 differently transcribed genes in the ΔCj1608 strain, 12 were also induced in AB$_S$ cells (Fig. S3D, red dots). Thus, these genes responded to oxidative stress but were controlled by factors other than or additional to Abu0127. KEGG enrichment analysis revealed that in the ΔAbu0127 strain, 2 KEGG groups were activated, ABC transporters and sulfur metabolism, while 13 groups were suppressed (e.g., oxidative phosphorylation and TCA cycle) (Fig. 2C). In the AB strain under stress conditions, seven KEGG groups were suppressed (e.g., oxidative phosphorylation and citrate cycle), while six groups were activated (e.g., sulfur metabolism).

As in our *C. jejuni* study, a proteomic approach was used to detect proteins whose levels changed between the strains and conditions analyzed. A total of 1,586 *A. butzleri* proteins were detected (S2 Data). However, proteomics did not detect 224 proteins encoded by genes whose transcription changed in ΔAbu0127 (S2 Data). The comparison between proteomes of ΔAbu0127 and AB strains under microaerobic conditions revealed 124 differentially expressed proteins (Fig. 2D; Fig. S4D, red, yellow, and green dots). As in *C. jejuni* studies, the transcriptomics and proteomics data correlated (Fig. 2D, red dots; Fig. S5B;); however, in cases of many genes, post-transcriptional modifications probably affected the final protein levels in the *A. butzleri* cell (Fig. 2D; blue, yellow, and green dots). Under paraquat stress, the level of only one protein changed; namely, Abu0530, of unknown function, was produced at a higher level in AB$_S$ cells compared to AB cells (Fig. S4E and F). Thus, *A. butzleri*, like *C. jejuni*, ceased protein translation upon oxidative stress, which is also typical for other bacterial species (37). The ClusterProfiler KEGG enrichment analysis revealed that genes of one KEGG group, sulfur metabolism, were activated in the ΔAbu0127 strain compared to AB. In contrast, the genes of three KEGG groups were suppressed (oxidative phosphorylation, taurine and hypotaurine metabolism, and ribosome) (Fig. 2E).

To conclude, the results of transcriptomic and proteomic analyses indicated that Abu0127 affects the transcription of 904 genes and the expression of 124 proteins in *A. butzleri* RM4018, with the oxidative phosphorylation KEGG group most affected.

## Cj1608 and Abu0127 are involved in the regulation of energy production and conversion

The transcriptome and proteome analyses of both species revealed that the energy production and conversion processes, such as the TCA cycle and oxidative phosphorylation, are putatively controlled in *C. jejuni* by Cj1608 and in *A. butlzeri* by Abu0127 (S1 Data, S2 Data and S3 Data). In *C. jejuni* ΔCj1608, the protein levels of all but two TCA enzymes, fumarate reductase FrdABC and fumarate hydratase FumC, decreased, with the citrate synthase GltA protein level reduced by as much as 70-fold (Fig. 3A; Fig. S6; the level of oxoglutarate-acceptor oxidoreductase Oor was reduced by 1.7- to 2.0-fold). Many genes encoding nutrient importers and downstream processing enzymes were downregulated in *C. jejuni* ΔCj1608 (e.g., amino acid ABC transporter permease Peb, fumarate importer DcuAB, L-lactate permease LctP, and lactate utilization proteins Lut). However, DctA succinate/aspartate importer and the enzymes glutamine synthetase GlnA and glutamate synthase small subunit GltB involved in glutamine and glutamate synthesis were upregulated (Fig. 3A). The levels of many proteins forming ETC complexes

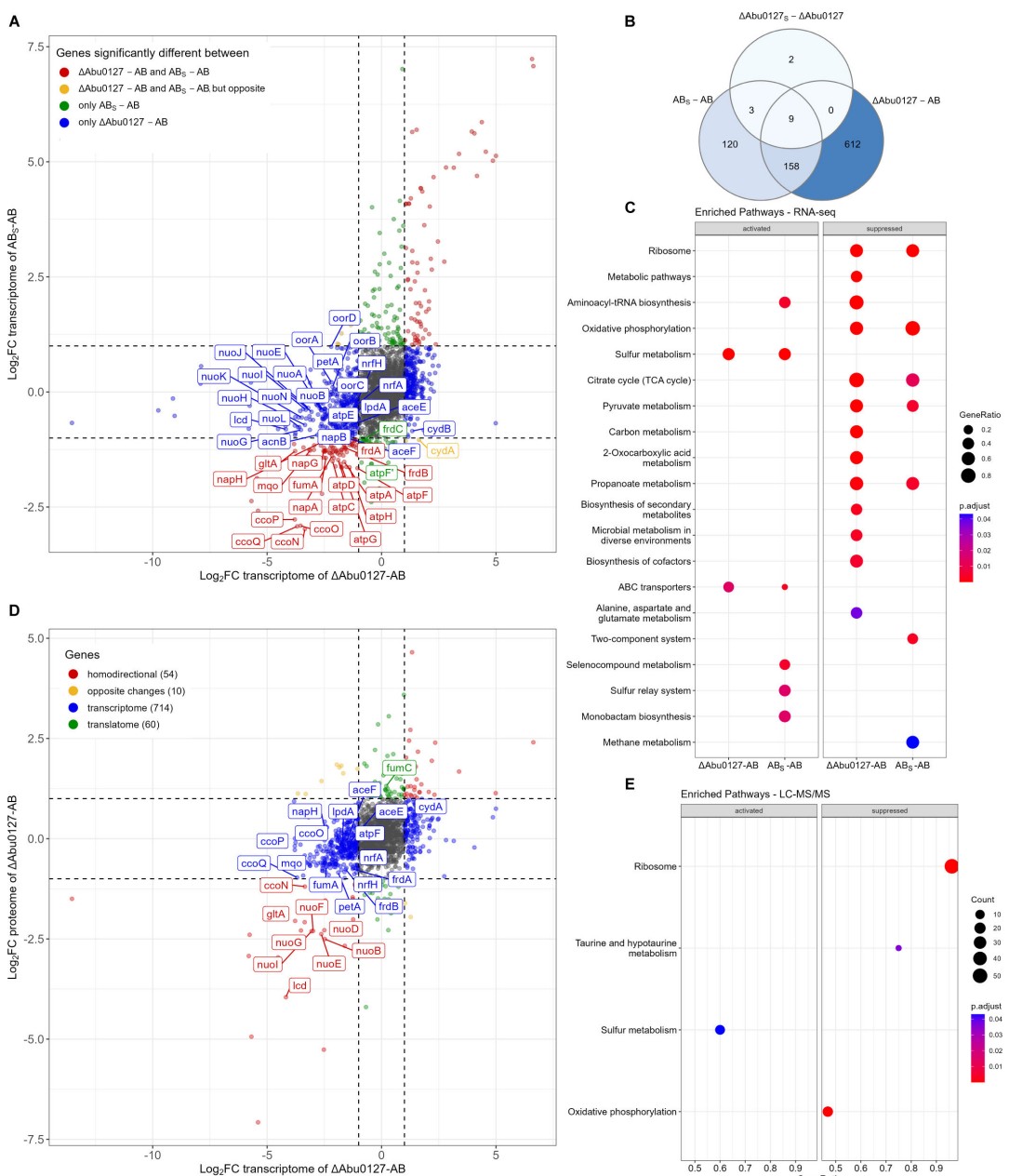

**FIG 2** RNA-seq and LC-MS/MS analyses of *A. butzleri* gene expression controlled by Abu0127. (A) The comparison of gene transcription in the AB, AB$_S$, and Abu0127 knock-out mutant (ΔAbu0127) cells revealed by RNA-seq. Genes whose transcription significantly changed ($|\log_2FC| \geq 1$; FDR < 0.05) are depicted by colored dots (see the legend in the figure). The genes named on the graph correspond to the citric acid cycle and electron transport chain. (B) The Venn diagram presents the number of differentially transcribed genes in the analyzed strains and conditions. (C) ClusterProfiler protein enrichment plot showing activation or suppression of KEGG groups in the analyzed strains. (D) The correlation between gene expression of ΔAbu0127 and AB cells at the transcription and proteome levels. Red dots represent homodirectional genes up-regulated or down-regulated at the transcriptome and proteome levels. Yellow dots represent opposite changes. Blue dots represent genes that changed only at the transcription level, while green dots represent genes that changed only at the proteome level. Numbers of differentially expressed genes in the indicated strains and conditions are given in parentheses. The proteins named on the graph correspond to the citric acid cycle and electron transport chain. (E) ClusterProfiler protein enrichment plot showing activation or suppression of KEGG groups in the analyzed strains. (A and D) Values outside the black dashed lines indicate a change in the expression of $|\log_2FC| \geq 1$. Gray dots correspond to genes whose transcription was not changed ($|\log_2FC| < 1$). AB, *A. butzleri* wild type; AB$_S$, stressed wild type, FC, fold change; FDR, false discovery rate; KEGG, Kyoto Encyclopedia of Genes and Genomes; LC-MS/MS, liquid chromatography coupled with tandem mass spectrometry.

were also downregulated, including subunits of NADH-quinone oxidoreductase Nuo and cytochromes (Fig. 3B). In *A. butzleri* ΔAbu0127, the levels of GltA and isocitrate

dehydrogenase Icd proteins were reduced (Fig. 3C; Fig. S7C). As in *C. jejuni*, the levels of many proteins forming ETC complexes decreased, with Nuo complex subunits being the most highly downregulated (Fig. 3D; Fig. S7A and B).

Moreover, a functional analysis annotation with eggNOG-mapper indicated the Clusters of Orthologous Groups (COGs) of differentially expressed genes and proteins. In both species, the most variable groups belonged to two categories, namely, energy production and conservation (C) and translation, ribosomal structure, and biogenesis (J) (Fig. 4; the category of unknown genes was excluded from the graph, S1 Data and S2 Data). Many genes and proteins suppressed in both species belong to the energy production and conversion group.

These comprehensive data suggest that mutants of both species produce less energy due to the inhibition of the TCA cycle and oxidative phosphorylation. Therefore, we decided to investigate these processes in more detail and analyze how the lack of Cj1608 and Abu0127 proteins influences energy production and conversion in *C. jejuni* and *A. butzleri*, respectively.

## Cj1608 controls *gltA* expression, ATP level, and growth in response to $O_2$ supply

To analyze the influence of Cj1608 on *C. jejuni* growth, we compared the growth of wild-type CJ, knock-out ΔCj1608, and complementation (C$_{Cj1608}$) strains (Fig. 5A). It should be noted that unless an accessory ETC electron donor and proton motive force generator $H_2$ was supplied in the microaerobic gas mixture (4% $H_2$; see Materials and Methods) [see also reference (38)], we could not obtain the ΔCj1608 mutant strain of *C. jejuni* NCTC 11168. When *C. jejuni* was cultured microaerobically in the presence of $H_2$, the growth of *C. jejuni* ΔCj1608 was slower [i.e., the generation time (G) was higher] than the CJ and C$_{Cj1608}$ strains (Fig. 5A). The ΔCj1608 culture entered the stationary phase at optical density (OD) like that of CJ and C$_{Cj1608}$ strains (Fig. 5A). Under micro-aerobic conditions without $H_2$, the CJ and C$_{Cj1608}$ strains grew similarly to conditions with $H_2$, albeit reaching lower $OD_{600}$ at the stationary phase than with $H_2$ (Fig. 5A). In contrast, the ΔCj1608 strain almost did not grow without $H_2$, confirming that its growth strictly depended on hydrogen. The ATP analysis indicated that the relative energy levels corresponded to the bacterial growth rates. The ATP level of the CJ strain under microaerobic growth without $H_2$ was assumed to be 100% (Fig. 5B). The ATP level in CJ and C$_{Cj1608}$ strains was constant, regardless of the presence or absence of $H_2$ (85%–100%). Without $H_2$, the ATP level in the ΔCj1608 strain dropped to 25% ± 4% of the ATP level in the CJ strain. In the presence of $H_2$, the level of ATP in ΔCj1608 cells increased to 53% ± 8% compared to that of the CJ strain. These results indicated that the availability of additional electrons and proton motive force derived from hydrogen (38), an alternative to those produced by the TCA cycle and used in oxidative phosphorylation, enabled ΔCj1608 cells to produce more energy and multiply more efficiently.

Next, we analyzed whether Cj1608 directly affects the TCA cycle efficiency, as suggested by hydrogen boost analysis. We studied the expression of *gltA* since the expression of this gene was severely downregulated in the ΔCj1608 strain (Fig. S8A and B; S3 Data). GltA is the first enzyme in the TCA cycle whose activity impacts the flow of substrates through the TCA cycle and energy production via NADH/FADH/menaqui-none cofactors (Fig. 3A). It was previously shown that *gltA* transcription depends on the $O_2$ concentration, being lower at 1.88% $O_2$ and higher at 7.5% $O_2$, which helps *C. jejuni* optimize energy production and expense during different oxygen availability as the electron acceptor (39). Here, we confirmed the specific interaction of the Cj1608 protein with the *gltA* promoter region *in vivo* by chromatin immunoprecipitation (ChIP) and *in vitro* by electrophoretic mobility shift assay (EMSA) (Fig. 5C and D); Cj1608 did not interact with a control *C. jejuni recA* region. Next, we used reverse transcription quantitative PCR (RT-qPCR) to analyze *gltA* transcription under oxidative stress triggered by paraquat or under diverse $O_2$ availability: reduced 1% $O_2$, optimal microaerobic 5% $O_2$, and increased 10% $O_2$ (see Materials and Methods). Transcription of *gltA* was lower

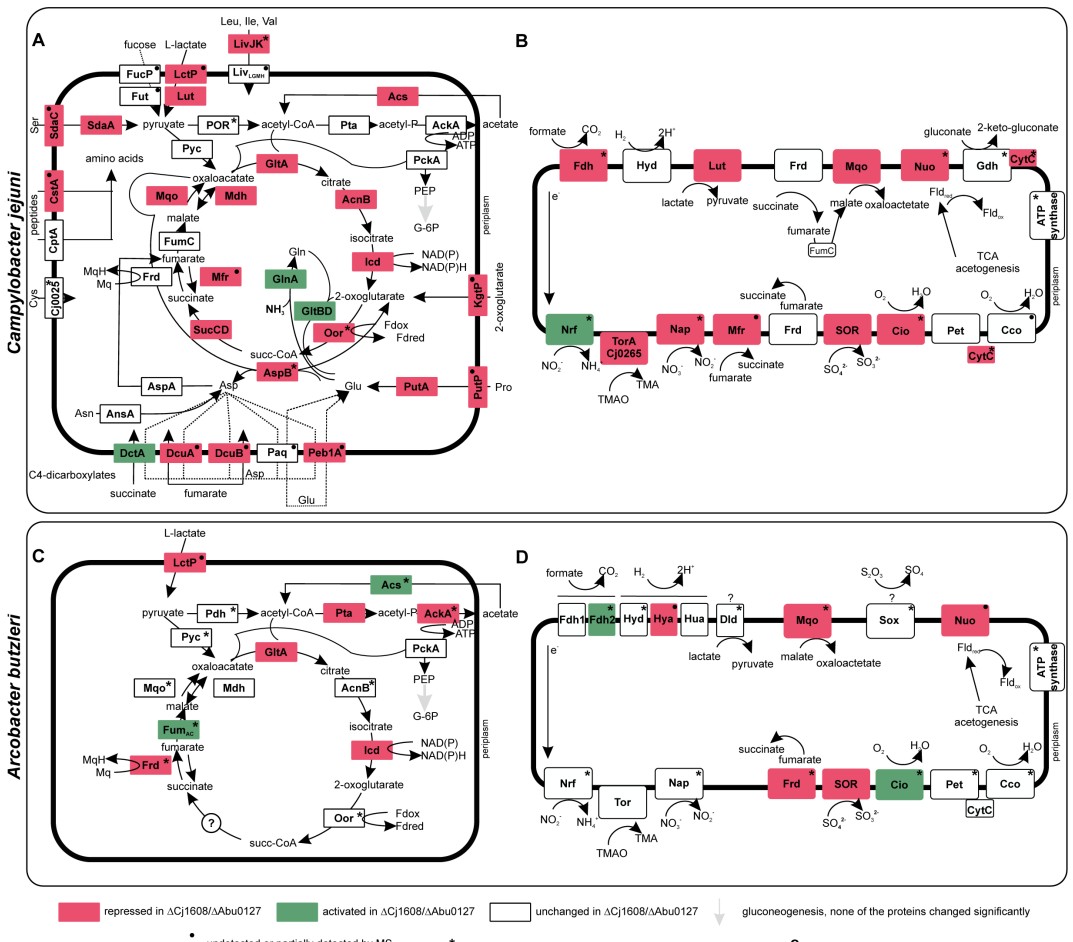

**FIG 3** Schematic presentation of *C. jejuni* and *A. butzleri* energy conservation factors whose expression changed in ΔCj1608 and ΔAbu0127, respectively. (A) Regulation of the *C. jejuni* citric acid cycle, the selected associated catabolic pathways, and nutrient transporters. (B) Regulation of *C. jejuni* electron transport chain enzymes. (C) Regulation of the *A. butzleri* tricarboxylic acid cycle selected associated catabolic pathways and nutrient transporters. (D) Regulation of *A. butleri* electron transport chain enzymes. (A–D) The increase or decrease in protein abundance in the deletion mutant strains compared to wild-type strains is shown based on the LC-MS/MS data (S3 Data). When MS did not detect the protein but the corresponding transcript level was increased or decreased, the color code depicts RNA-seq results (denoted as •). For details concerning transcriptomic data or complex transcriptome/proteome interpretation (denoted as *), see S1 to S3 Data. Cco, cbb3-type cytochrome c oxidase; Cio, cyanide-insensitive cytochrome bd-like quinol oxidase; Fdh, formate dehydrogenase; Frd, fumarate reductase; G-6P, glucose 6-phosphate; Gdh, gluconate 2-dehydrogenase; Hyd, Hya, and Hua, hydrogenases; Lut, Dld, L-lactate utilization complex; Mfr, methylmenaquinol:fumarate reductase (misannotated as succinate dehydrogenase Sdh); Mqo, malate:quinone oxidoreductase; Nap, nitrate reductase; Nrf, nitrate reductase; Nuo, NADH-quinone oxidoreductase; Oor, 2-oxoglutarate:acceptor oxidoreductase; PEP, phosphoeneoopyruvate; Pet, ubiquinol-cytochrome C reductase; POR, pyruvate:acceptor oxidoreductase; SOR, sulfite:cytochrome c oxidoreductase; SOX, thiosulfate oxidation by SOX system; Tor, SN oxide reductase; "?," unknown enzyme. All genes and enzyme complexes can be found in S3 Data.

in ΔCj1608 than in CJ and $C_{Cj1608}$ strains under optimal conditions [fold change (FC) of 0.14 ± 0.14] and unaffected by paraquat stress, confirming the transcriptomic results (Fig. S8C). Moreover, transcription of *gltA* was upregulated in the CJ and $C_{Cj1608}$ strains at 5% $O_2$ compared to 1% $O_2$ (Fig. 5E). However, in the CJ and $C_{Cj1608}$ strains, the *gltA* transcription was downregulated at 10% $O_2$ compared to 5% $O_2$ (Fig. 5E), suggesting that the prolonged bacterial growth for 5 h at increased $O_2$ concentration possibly caused adaptation to higher oxygen level conditions or stress by reducing TCA cycle activity and possible ROS production during increased aerobiosis. However, *gltA* transcription was constant in ΔCj1608 at all $O_2$ concentrations, indicating that cells could not adjust *gltA* transcription to changing $O_2$ conditions. Next, we analyzed *C. jejuni* growth and ATP

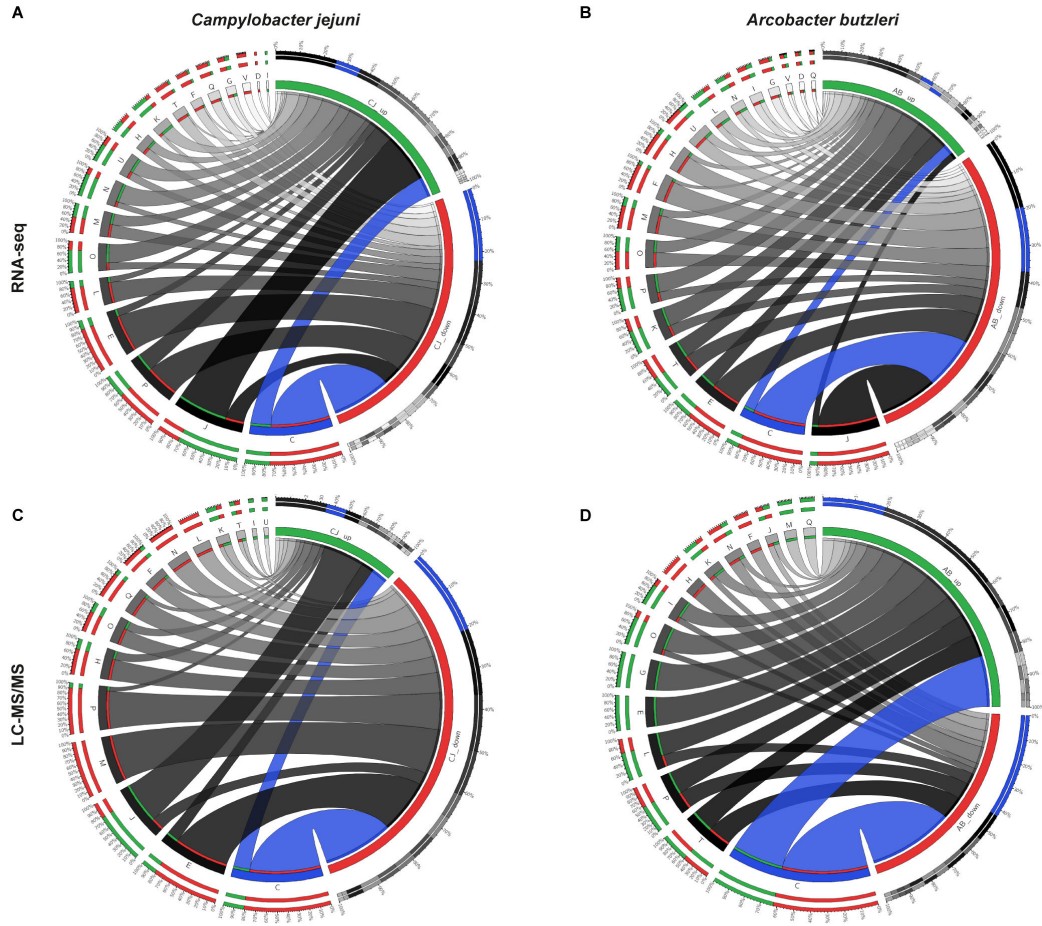

**FIG 4** Impact of Cj1608 and Abu0127 on gene regulation in *C. jejuni* and *A. butzleri*, respectively. (A) Circos plots presenting the correlation between COG and differentially expressed *C. jejuni* genes revealed by RNA-seq in the ΔCj1608 strain compared to the *C. jejuni* wild-type strain. (B) Circos plots presenting the correlation between COG and differentially expressed *A. butzleri* genes revealed by RNA-seq in the ΔAbu0127 strain compared to the *A. butzleri* wild-type strain. (C) Circos plots presenting the correlation between COG and differentially expressed *C. jejuni* proteins revealed by LC-MS/MS in the ΔCj1608 strain compared to the *C. jejuni* wild-type strain. (D) Circos plots presenting the correlation between COG and differentially expressed *A. butzleri* proteins revealed by LC-MS/MS in the ΔAbu0127 strain compared to the *A. butzleri* wild-type strain. (A–D) Expression changes of |log2FC| ≥ 1 and FDR < 0.05 were considered significant and were included in the analyses. The category of unknown genes was excluded from the analysis. COG, Cluster of Orthologous Groups.

**C** - energy production and conversion, **D** - cell cycle control, cell division, chromosome partitioning, **E** - amino acid transport and metabolism, **F** - nucleotide transport and metabolism, **G** - carbohydrate transport and metabolism, **H** - coenzyme transport and metabolism, **I** - lipid transport and metabolism, **J** - translation, ribosomal structure and biogenesis, **K** - transcription, **L** - replication, recombination and repair, **M** - cell wall/membrane/envelope biogenesis, **N** - cell motility, **O** - posttranslational modification, protein turnover, chaperones , **P** - inorganic ion transport and metabolism, **Q** - secondary metabolites biosynthesis, transport and catabolism, **T** - signal transduction mechanisms, **U** - intracellular trafficking, secretion, and vesicular transport, **V** - defense mechanisms

production under various $O_2$ supplies. Under 1% $O_2$, the ΔCj1608 strain grew faster than WT and $C_{Cj1608}$ strains, suggesting that the metabolic state of the ΔCj1608 mutant strain was optimized for reduced $O_2$ concentration (Fig. 5G). Under optimal $O_2$ level, CJ and $C_{Cj1608}$ strains grew faster than under 1% $O_2$ and faster than the ΔCj1608 strain under 5% $O_2$, indicating that metabolic pathways of CJ and $C_{Cj1608}$ strains were adjusted to increased $O_2$ availability, while the metabolism of ΔCj1608 was not. Under increased $O_2$ concentration, all three strains grew slower than under optimal conditions, with the ΔCj1608 strain growing slower than the CJ and $C_{Cj1608}$ strains. The ATP analyses indicated that the relative ATP levels corresponded to the bacterial growth rates. The ATP level of the CJ strain under optimal microaerobic growth was assumed to be 100%. Compared to that, the level of ATP in the CJ strain under reduced $O_2$ concentration dropped to 38% ± 4%; at increased $O_2$ concentration, the level of ATP also decreased to 51% ± 7% (Fig. 5F; see Discussion). Under optimal conditions in the ΔCj1608 mutant

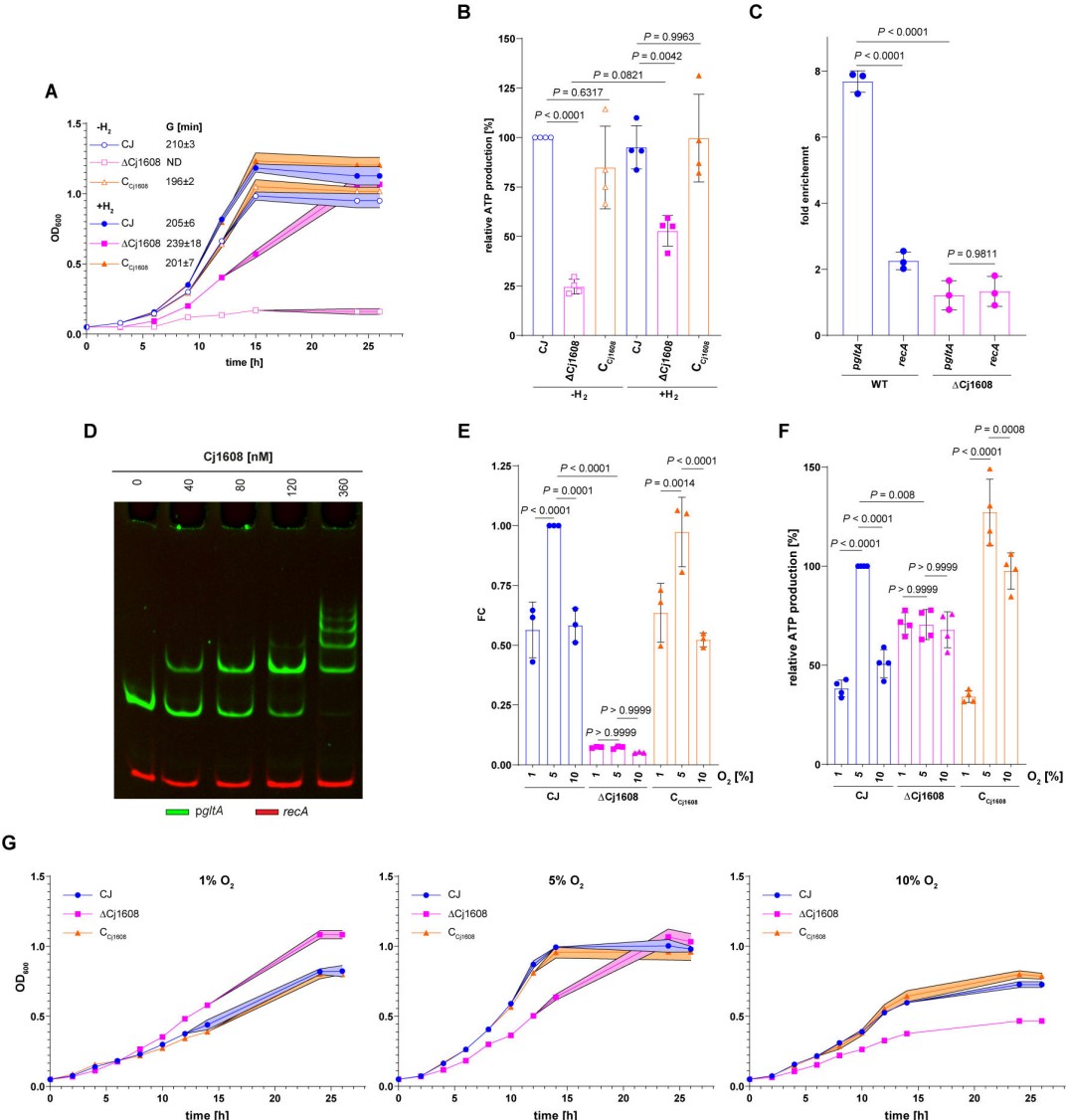

**FIG 5** Cj1608-dependent control of *C. jejuni* energy conservation pathways. (A) Growth curves and generation times of CJ, ΔCj1608, and C$_{Cj1608}$ strains cultivated in the presence of H$_2$ or without H$_2$. (B), ATP production by CJ, ΔCj1608, and C$_{Cj1608}$ strains cultivated as in part A. (C) ChIP-qPCR analysis of p*gltA* fragment immunoprecipitated from CJ and ΔCj1608 cultured under microaerobic conditions; the *recA* gene was used as a negative control which Cj1608 does not bind. (D) EMSA analysis of Cj1608 binding to the p*gltA*-FAM promoter region *in vitro*; the *recA*-Cy5 fragment was used as a negative control. (E) RT-qPCR analysis of *gltA* transcription in CJ, ΔCj1608, and C$_{Cj1608}$ cells cultured at different oxygen concentrations: (i) 1% O$_2$, (ii) 5% O$_2$, or (iii) 10% O$_2$ (see Materials and Methods). (F) Growth curves of CJ, ΔCj1608, and C$_{Cj1608}$ strains cultivated as in panel E. (G) ATP production by CJ, ΔCj1608, and C$_{Cj1608}$ strains cultivated as in panel E. (A–C and E–G) Data presented as the mean values ± SD. Ordinary one-way ANOVA with Tukey's multiple comparison test determined the *P* value. *n* = 3 biologically independent experiments. ANOVA, analysis of variance; ChIP-qPCR, chromatine immunoprecipitation quantitive PCR; CJ, *C. jejuni* wild type; C$_{Cj1608}$, Cj1608 complementation mutant; ΔCj1608, Cj1608 knock-out mutant ; EMSA, Electrophoretic mobility shift assay; G, generation time; ND, non-determined; RT-qPCR, Reverse transcription quantitative PCR.

strain, the ATP level reached approximately 70% ± 8% of that in the CJ strain and was constant regardless of increased or decreased O$_2$ supply. The ATP levels in the C$_{Cj1608}$ strain analyzed at different O$_2$ levels resembled that of the CJ strain. Thus, the results indicated that the ΔCj1608 mutant strain could not adjust pathways responsible for energy conservation to changing O$_2$ levels.

To summarize *C. jejuni*'s results, Cj1608 helps *C. jejuni* control pathways that are important for energy conservation and are dependent on oxygen availability. As we have

shown using the *gltA* gene as an example, Cj1608 directly controls gene expression in response to changing $O_2$ concentrations and oxidative stress.

## Abu0127 controls *nuo* expression, ATP level, and growth in response to $O_2$ supply

Next, we examined the impact of Abu0127 on *A. butzleri's* growth. As for *C. jejuni*, the presence or absence of $H_2$ did not significantly affect the growth of the wild-type AB strain nor ATP level in AB cells under microaerobic conditions (Fig. 6A). In the presence of $H_2$, the ΔAbu0127 strain grew slower than the AB strain (Fig. 6A), and the ATP level of ΔAbu0127 cells reached 77% ± 3% of that in the WT cells (Fig. 6B). However, the ΔAbu0127 culture entered the stationary phase at $OD_{600}$ comparable to that of the AB strain. Under microaerobic conditions without $H_2$, the growth and ATP level of ΔAbu0127 was further lowered; ΔAbu0127 ATP level reached 57% ± 3% of the AB cells, and cells entered stationary phase at lower $OD_{600}$ than the wild-type AB strain (Fig. 6A and B). Despite many attempts, we could not construct an Abu0127 complementation mutant, possibly due to the imperfect molecular biology tools dedicated to *A. butzleri* (10). Nonetheless, the results indicated that the ΔAbu0127 strain produced less energy than AB cells, and using $H_2$, the ΔAbu0127 cells could produce more energy and multiply more efficiently (38). This suggests a similar pattern of bypass of TCA-dependent energy production via oxygen-dependent oxidative phosphorylation to that observed in *C. jejuni*.

Next, we analyzed whether Abu0127 directly affects the Nuo complex activity. We studied the expression of *nuoB* since it is the second gene of the *nuoA-N* operon, whose expression was severely downregulated in the ΔAbu0127 strain at the transcription and translation levels (Fig. S9A and B; S3 Data). We confirmed the interaction of the Abu0127 protein with the *nuoA-N* promoter region *in vivo* by ChIP-qPCR and *in vitro* by EMSA (Fig. 6C and D), and we found that it was specific because Abu0127 did not interact with control *A. butzleri gyrA* and *recJ* regions, respectively. Next, we used RT-qPCR to analyze *nuoB* transcription under paraquat-induced oxidative stress and different $O_2$ supply. It should be noted that *A. butzleri* can grow under aerobic conditions (40); thus, aerobic $O_2$ concentration is less harmful to *A. butzleri* than to *C. jejuni*. The transcription of *nuoB* changed across different $O_2$ concentrations, being the lowest at 1% $O_2$ (FC of 0.61 ± 0.06 compared to AB under 5% $O_2$) and highest at 10% $O_2$ (FC of 1.7 ± 0.27 compared to AB under 5% $O_2$) (Fig. 6E). Under optimal $O_2$ conditions, *nuoB* transcription was lower in the ΔAbu0127 strain than in the AB strain (FC of 0.3 ± 0.05 compared to AB under 5% $O_2$), and it was invariant across different $O_2$ concentrations (Fig. 6H). Paraquat-induced oxidative stress affected *nuoB* transcription neither in the wild-type AB nor ΔAbu0127 strain, confirming the transcriptomic results (Fig. S9C, S3 Data). Next, we analyzed *A. butzleri* growth and ATP production under various $O_2$ supplies. Under 1% $O_2$, the AB and ΔAbu0127 strains grew similarly. However, ΔAbu0127 reached a lower $OD_{600}$ at the stationary growth phase than the AB strain (Fig. 6F). Under optimal and increased $O_2$ concentrations, the AB strain grew similarly at both concentrations, faster than under 1% $O_2$, but reached a similar $OD_{600}$ upon entry to a stationary growth phase as under 1% $O_2$. The ΔAbu0127 strain grew similarly under 5% and 10% $O_2$ but faster than under 1% $O_2$. Nonetheless, ΔAbu0127 grew slower than AB under the same conditions, and the culture finally reached a lower $OD_{600}$ than AB (Fig. 6F). The relative ATP energy levels corresponded to the bacterial growth rates. The ATP level of the AB strain under microaerobic growth was assumed to be 100%. Compared to that, under 1% $O_2$, the level of ATP in the AB strain dropped to 44% ± 4%, while the level of ATP did not change in the culture grown at 10% $O_2$ (Fig. 6G). Under reduced $O_2$ concentration, the level of ATP in ΔAbu0127 was similar to that of the AB strain under the same conditions (35% ± 4% compared to AB under microaerobic growth). However, under 5% $O_2$, the ATP level of the ΔAbu0127 mutant strain reached 76% ± 3% of that in the AB strain and did not increase under 10% $O_2$. Thus, the results indicated that the ΔAbu0127 mutant strain could not

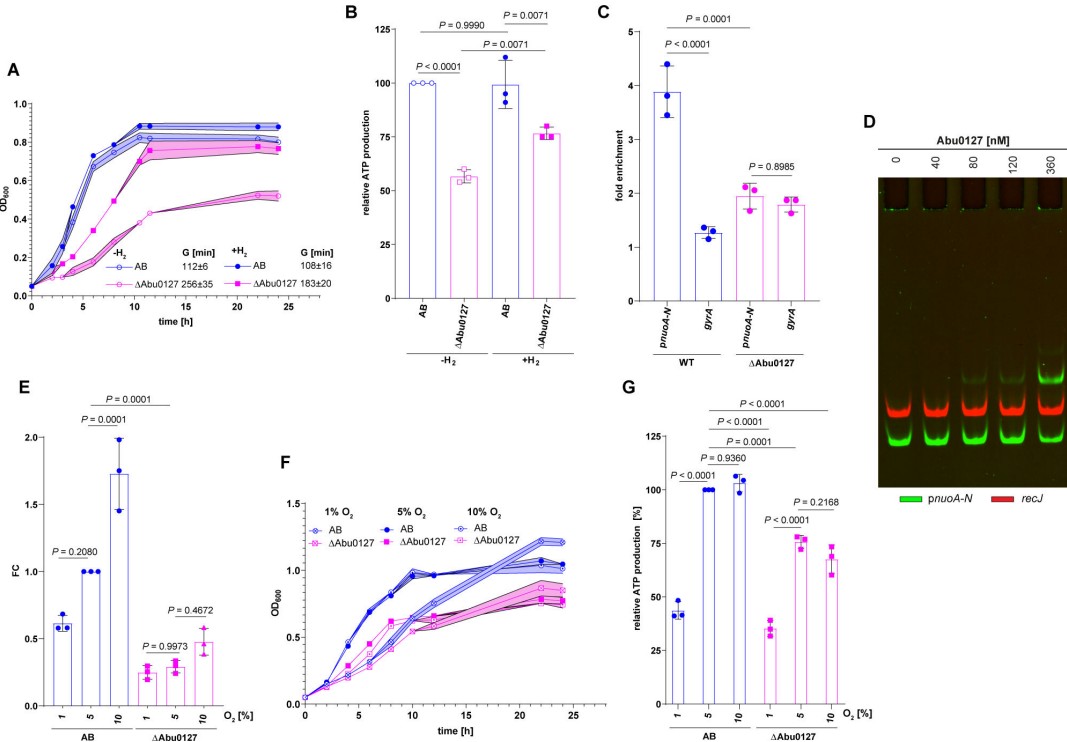

**FIG 6** Abu0127-dependent control of *A. butzleri* energy conservation pathways. (A) Growth curves and generation times of AB and ΔAbu0127 strains cultivated in the presence of H$_2$ or without H$_2$. (B) ATP production by AB and ΔAbu0127 strains cultivated as in panel A. (C) ChIP-qPCR fold enrichment of DNA fragment in p*nuoA-N* by ChIP-qPCR in AB and ΔAbu0127 cultured under microaerobic conditions. The *gyrA* gene was used as a negative control not bound by Abu0127. (D) EMSA analysis of Abu0127 binding to the p*nuoA*-FAM promoter region *in vitro*; the *recJ*-Cy5 fragment was used as a negative control. (E) RT-qPCR analysis of *nuoB* transcription in AB and ΔAbu0127 strains cultured at different oxygen concentrations: (i) 1% O$_2$, (ii) 5% O$_2$, or (iii) 10% O$_2$ (see Materials and Methods). (F) Growth curves of AB and ΔAbu0127 strains cultivated as in panel E. (G) ATP production by AB and ΔAbu0127 strains cultivated as in panel E. (A–C, E–G) Data presented as the mean values ± SD. Ordinary one-way ANOVA with Tukey's multiple comparison test determined the *P* value. *n* = 3 biologically independent experiments. AB, *A. butzleri* wild type; ANOVA, analysis of variance; ΔAbu0127, Abu0127 knock-out mutant; ChIP, chromatin immunoprecipitation quantitative PCR; EMSA, electrophoretic mobility shift assay; G, generation time; RT-qPCR, reverse transcription quantitative PCR.

efficiently adjust pathways responsible for energy conservation to changing O$_2$ levels, in which it resembled *C. jejuni* ΔCj1608.

To summarize *A. butzleri*'s results, Abu0127 helps *A. butzleri* control pathways important for energy conservation dependent on oxygen availability. As we have shown using the *nuoB* gene as an example, Abu0127 directly controls gene expression in response to changing O$_2$ concentrations.

## DISCUSSION

One of the challenges for a bacterial cell is to gain energy to grow and reproduce under different environmental conditions. Available energy sources and electron acceptors used for energy conversion are the two major factors defining the activity of metabolic pathways. The regulatory proteins redirect metabolism and ETC pathways, prioritizing electron donor or acceptor usage to maximize the energy gain (e.g., aerobic/microaerophilic over anaerobic respiration and fermentation or nitrate respiration over fumarate respiration). The fine-tuning of metabolism and respiration to particular conditions is usually hierarchical and orchestrated by multiple factors (41, 42), which often control the same genes. For example, in the model, facultative anaerobe *Escherichia coli*, but also other species of *Gammaproteobacteria*, three global transcriptional regulators, FNR, ArcA, and NarL/NarP, mainly control energy conservation processes dependent on the availability of electron acceptors (21, 43). In the energy reprogramming network, under low oxygen availability, FNR provides the first level of regulation, switching metabolism

from aerobic to anaerobic and controlling the expression of other transcriptional factors (44). ArcA, a response regulator of the ArcAB two-component system (45), possibly controls the switch between anaerobic respiration and fermentation, as well as the expression of a set of secondary regulators (46). NarL/NarP switches on nitrate and nitrite respiration when these electron acceptors are available while repressing genes for less effective anaerobic respiration (47). Studies have shown that the intensive cross-talk between regulatory proteins, often controlling the same genes hierarchically and controlling genes beyond energy conservation pathways, is essential for efficient bacterial responses or adaptation to diverse conditions (21, 48–50).

Until recently, no regulators redirecting energy conversion pathways in response to oxygen as an electron acceptor have been found in *Campylobacterota* (14, 51). It was experimentally shown that *C. jejuni* reprograms its metabolism upon changes to oxygen availability; nonetheless, the regulator remained unknown (39, 52). It was surprising, given that microaerobic species of this phylum must respond to changing oxygen levels, including harmful oxygen deficiency and excess.

In this and previous works (24, 26), we revealed that under microaerobic conditions, three homologous atypical response regulators, HP1021, Cj1608, and Abu0127, which we name *Campylobacteria* energy and metabolism regulators (CemR), control the expression of many genes involved in energy conversion, including TCA and ETC pathways [Fig. 3 and 4; Fig. S10; Supplementary Data 3 in reference (26)]. Indeed, when *cemR* genes were deleted, the most remarkable gene and protein expression changes occurred in energy production and conversion COG, reaching between 20% and 40% of all the downregulated genes or proteins (Fig. 4; Fig. S10; the category of unknown proteins was excluded from the study). Moreover, we showed that in strains lacking CemR, the transcriptional control was lost over the genes whose transcription was dependent on the oxygen level, e.g., *C. jejuni* TCA *gltA*, *A. butzleri* ETC, *nuoA-N* operon complex, *H. pylori*, TCA pyruvate:ferredoxin oxidoreductase *PFOR*, and ETC cytochrome c oxidase *ccoN-Q* [Fig. 5E and 6E and reference (26)]. Consequently, under microaerobic conditions in CemR deletion mutants of all three species, the ATP level and the growth rates were lower than in wild-type strains [Fig. 5F, G and 6F G, , and (26)]. The presence of CemR allowed *C. jejuni* and *A. butlzeri* to optimize energy conservation under microaerobic compared to reduced oxygen concentration, increasing the ATP level and bacterial growth rate (Fig. 5F, G and 6F G, ). The lack of CemR specifically impaired pathways connected with oxygen utilization as an electron acceptor because in the presence of hydrogen as an alternative electron donor and proton motive force generator, *C. jejuni* and *A. butzleri* gained additional energy, allowing for faster growth (Fig. 5A, B and 6A B, ). All these data indicate that CemR responds to oxygen levels and redirects metabolism toward optimal energy conservation. It is important to point out that data obtained from each single species would not have allowed us to conclude on the general role of CemR as an energy conservation regulator because, in each species, slightly different pathways were primarily controlled (e.g., TCA in *C. jejuni*, Nuo ETC in *A. butzleri*, and glucose uptake and pentose phosphate pathway in *H. pylori*). Nonetheless, the role of CemR in controlling other pathways or processes, often in a species-specific manner, still awaits detailed characterization.

An interesting question arises: how many energy conservation regulators are encoded in addition to CemR in each *Campylobacteria* species, and how do they all integrate into the regulatory circuits of a given species? *H. pylori*, *C. jejuni*, and *A. butzleri* differ in lifestyles, which shape the complexity of regulatory circuits, as recently illustrated using *Campylobacterota* as a representative phylum (20). *H. pylori* can be classified as a specialist bacterium (53), strictly human associated, using a limited repertoire of electron donors (hydrogen, pyruvate, 2-oxo-glutarate, malate, and succinate), and only oxygen or fumarate as electron acceptors during respiration (20). Oxygen, being a primary electron acceptor in *H. pylori*, is also toxic for *H. pylori* at higher than microaerobic concentrations (54). As a consequence of speciation and host adaptation, *H. pylori* lost many regulatory proteins. Nonetheless, the studies on

regulatory circuits in *H. pylori* indicated that HP1043 (55, 56), controlled by unknown stimuli, acid-responsive ArsRS (57), and metal-dependent regulators NikR and Fur (20, 58), affects *H. pylori* energy conversion to various extents in response to different stimuli. *C. jejuni* is a host-associated organism that infects various animals and humans and can survive long periods under aerobic conditions (59). In contrast to *H. pylori*, *C. jejuni* has a branched electron transport chain with multiple enzymes that utilize more molecules as electron donors and acceptors than *H. pylori* (20). *C. jejuni* genome size is similar to *H. pylori*; however, the number of *C. jejuni* genes encoding regulatory proteins is higher and reflects the broader spectrum of ecological niches inhabited by this bacterium. Consequently, the number of regulatory proteins controlling energy conversion is higher than that in *H. pylori*, with RacRS, LysR (Cj1000), CsrA, and CprRS playing the most significant roles [for details, see reviews (11, 13–15)]. Depending on growth conditions, particularly the availability of electron donors and acceptors, LysR, RacRS, and CprRS regulate the expression of fumarate respiration genes, which is an excellent example of a cross-talk between regulators controlling the same pathways via transcriptional control over the same genes (e.g., *aspA*, *dcuA*, *mfr*, or *frd*) (51, 60, 61). RNA-binding CsrA, a homolog of *E. coli* carbon starvation regulator, is an example of post-transcriptional energy conservation regulation in *C. jejuni*, including control of expression of TCA and ETC genes (62). *A. butzleri* can lead anaerobic, microaerophilic, and aerobic lifestyles as an environmental or animal- and human-associated bacterium. As a generalist, it utilizes many electron donors and acceptors using an electron transport chain similar to those used by the free-living marine *Campylobacterota* (20). *A. butzleri*'s genome encodes many signal transduction proteins (see Introduction), including extracytoplasmic sigma factors (ECF) to adapt to different conditions. However, hardly anything is known about the regulation of *A. butzleri* energy conversion except that a few species of ECFσ impact electron and carbon metabolism by affecting the transcription of genes from carbon metabolism pathways and electron acceptor complexes (10). However, one can expect the highest complexity and speciation of *A. butzleri*'s regulatory circuits, including energy conservation, of all three species.

CemR regulators are highly conserved *Campylobacteria* regulators. However, the molecular mechanism of signal perception by CemR is still enigmatic and cannot be predicted by analogy. Transcriptional regulatory proteins may sense respiration status, the concentrations of electron donors or acceptors, or respiration byproducts (63). We have shown that cysteine residues of HpCemR become oxidized under $O_2$-triggered oxidative stress *in vivo* and *in vitro*, which changes the protein's DNA-binding activity possibly by structural changes of DNA-protein complex rather than on-off mechanism (24, 26). Recent studies on CjCemR interactions with the promoter region of *lctP* suggested a similar mechanism of CjCemR activity control and *lctP* transcription regulation (27). AbCemR protein also contains cysteine residues, one of which is conserved in all three species (C27 in HpCemR and CjCemR, C31 in AbCemR). The mechanism of activity regulation by the redox state of the cysteine residues resembles that of ArcA-ArcB two-component system regulation. However, in that TCS, the cysteine residues of ArcB sensor kinase are oxidized, triggering autophosphorylation of ArcB, while ArcA response regulator is activated by phosphorylation. CemR proteins are orphan atypical response regulators in which the same molecule receives and executes the signal. Thus, despite some biochemical similarity to ArcAB system activation, the exact molecular mechanism of CemR activity control has not yet been discovered.

In summary, CemR, the global regulatory protein, is a part of the cell response to a metabolic redox imbalance either caused by environmental conditions such as oxygen availability or ROS or triggered intracellularly due to metabolic changes and increased ROS production. However, the exact pathway of signal transduction by CemR and the levels or hierarchy of regulation are still unknown. Further comprehensive, multi-omic studies are needed to reveal the complex circuits of regulation involving CemR and other regulatory proteins in *Campylobacteria* energy conservation.

## MATERIALS AND METHODS

### Materials and culture conditions

The strains, plasmids, and proteins used in this work are listed in the Table S1. The primers used in this study are listed in Table S2. *C. jejuni* and *A. butzleri* plate cultures were grown on Columbia blood agar base medium supplemented with 5% defibrinated sheep blood (CBA-B). The liquid cultures were prepared in brain heart infusion broth (BHI) (Oxoid) and incubated with 140-rpm orbital shaking. All *C. jejuni* and *A. butzleri* cultures were supplemented with antibiotic mixes [*C. jejuni*: vancomycin (5 µg/mL), polymyxin B (2.5 U/mL), trimethoprim (5 µg/mL), and amphotericin B (4 µg/mL) (64); *A. butzleri*: cefoperazone (8 µg/mL), amphotericin (10 µg/mL), and teicoplanin (4 µg/mL) (65)]. If necessary for selecting mutants, appropriate antibiotics were used with the following final concentrations: (i) kanamycin 20 µg/mL (*C. jejuni* and *A. butzleri*) and (ii) chloramphenicol 8 µg/mL (*C. jejuni*). Routinely, *C. jejuni* and *A. butzleri* were cultivated at 42°C or 30°C, respectively, under optimal microaerobic conditions (*C. jejuni*: 5% $O_2$, 8% $CO_2$, 4% $H_2$, and 83% $N_2$; *A. butzleri*: 6% $O_2$, 9% $CO_2$, and 85% $N_2$) generated by the jar evacuation-replacement method using Anaerobic Gas System PetriSphere. The gas mixtures were modified by lowering or increasing $O_2$ and $H_2$ concentrations for ATP assays and growth curve analyses. Briefly, bacteria were cultured under microaerobic conditions optimal for each species to the late logarithmic growth phase ($OD_{600}$ ~0.8), diluted to an $OD_{600}$ of ~0.05 and incubated under the desired atmosphere as long as required, up to $OD_{600}$ = 0.2–0.6 for ATP and RT-qPCR or until late stationary phase in growth analyses. The gas mixture with or without $H_2$ was composed of optimal gas mixtures for *C. jejuni* and *A. butzleri*, respectively. Low oxygen gas mixture was composed of 1% $O_2$, 10% $CO_2$, 5% $H_2$, and 84% $N_2$, while high oxygen gas mixture was composed of 10% $O_2$, 5% $CO_2$, 3% $H_2$, and 82% $N_2$. *Escherichia coli* DH5α and BL21 were used for cloning and recombinant protein synthesis. If necessary for selecting *E. coli*, appropriate antibiotics were used with he following final concentrations: (i) kanamycin 50 µg/mL and (ii) ampicillin 100 µg/mL.

### Construction of *C. jejuni* mutant strains

*C. jejuni* mutant strains were constructed using a homologous recombination and natural transformation approach (66). *C. jejuni* NCTC 11168 was grown in 12-mL BHI to an $OD_{600}$ = 0.2. Next, 150 µL of the culture was centrifuged and resuspended in fresh 150 µL of BHI. One hundred fifty nanograms (1 µg/mL) of EcoRI methylated plasmid was added to the culture and cultivated at 42°C in microaerobic (5% $O_2$, 8% $CO_2$, 4% $H_2$, and 83% $N_2$) conditions with shaking (140 rpm) for 4 h. Next, 100 µL was spread on CBA-B plates with an appropriate antibiotic and incubated for 3 days at 42°C under microaerobic (5% $O_2$, 8% $CO_2$, 4% $H_2$, and 83% $N_2$) conditions.

### *C. jejuni NCTC 11168 ΔCj1608*

The *C. jejuni Cj1608* deletion construct (pCR2.1/ΔCj1608) was prepared as follows (Fig. S11A). The upstream and downstream regions of *Cj1608* were amplified by PCR using the P1-P2 and P5-P6 primer pairs, respectively, and a *C. jejuni* NCTC 11168 genomic DNA as a template. The *aphA-3* cassette was amplified using the P3-P4 primer pair; pTZ57R/TΔHP1021 (23) was used as a template. The resulting fragments were purified on an agarose gel. Subsequently, the fragments were combined into one DNA amplicon with an overlap extension PCR reaction using the P1-P6 primers. The generated amplicon was purified on an agarose gel and cloned to the pCR2.1-TOPO plasmid (Thermo Fisher Scientific) according to the manufacturer's protocol. The DNA fragment cloned in pCR2.1-TOPO was sequenced. Subsequently, *C. jejuni* NCTC 11168 was transformed with the pCR2.1/ΔCj1608 plasmid, and the transformants were selected by plating on CBA-B plates supplemented with kanamycin. The addition of hydrogen was necessary to obtain the kanamycin-resistant colonies. The allelic exchange was verified by PCR using

the P7-P8 primer pair. The lack of *Cj1608 in C. jejuni* NCTC 11168 ΔCj1608 was confirmed by RNA-seq (Fig. S11B), LC-MS/MS (S1 Data), and Western blot (Fig. S11C and D).

## C. jejuni NCTC 11168 C$_{Cj1608}$

The *C. jejuni Cj1608* complementation construct (pUC18/COMCj1608) was prepared as follows (Fig. S11A). The regions upstream and downstream of the *Cj1608* gene were amplified by PCR using the P1-P6 primer pair and *C. jejuni* Cj1608 genomic DNA as a template. The resulting fragment was purified on an agarose gel. Subsequently, the generated amplicon and the SmaI digested vector pUC18 were ligated according to the method described by Gibson et al. (67) to give pUC18/COMCj1608. *E. coli* DH5α competent cells were transformed with the construct by heat shock. The insert cloned into pUC18 was sequenced. The pUC18/COMCj1608 was used to complement the lack of *Cj1608* in *C. jejuni* NCTC 11168 ΔCj1608 by an attempt similar to Multiplex Genome Editing by Natural Transformation (MuGENT) (68). For the selection, a pSB3021 suicide plasmid was used. The plasmid harbors a *cat* cassette flanked by arms of homology, enabling *cat* cassette integration into the *hsdM* gene. Subsequently, the obtained pUC18/COMCj1608 (1 µg/mL) and pSB3021 (0.25 µg/mL) plasmids were used in the natural transformation of *C. jejuni* NCTC 11168 ΔCj1608. The transformants were selected by plating on CBA-B plates supplemented with chloramphenicol. The bacteria were cultured without hydrogen to increase the selection process. The allelic exchange was verified by PCR using the P7-P9 primers. The presence of *Cj1608* in *C. jejuni* NCTC 11168 C$_{Cj1608}$ was confirmed by RNA-seq (Fig. S11B), LC-MS/MS (S1 Data), and Western blot (Fig. S10C and D).

## Construction of *A. butzleri* mutant strain

*A. butzleri* RM4018 mutant strain was constructed using a homologous recombination and electroporation approach. *A. butzleri* was grown in 25-mL BHI to an OD$_{600}$ = 0.4. Next, the culture was transferred on ice and left for 10 min, then centrifuged (4,700 g, 10 min, 4°C) and washed twice with 20 mL of ice-cold glycerine water (15% glycerol and 7% sucrose, filtrated). Finally, the culture was suspended in 250 µL of ice-cold glycerine water. For electroporation, 50 µL of electrocompetent cells was mixed with 5 µg of appropriate plasmid. Electroporation was performed in 0.1-cm electroporation cuvettes (The Cell Projects) using a Gene Pulser II Electroporator (Bio-Rad) using the following parameters: 12.5 kV/cm, 200 Ω, and 25 µF. Cells were regenerated by adding 1 mL of BHI medium to the cuvette, then by transferring the cells to a flask with 2-mL BHI and cultivated with shaking (140 rpm) for 4 h at 30°C in microaerobic conditions. Next, the culture was centrifuged (4,700 g, 10 min, 22°C), resuspended in 100 µL of BHI, spread on CBA-B plates with appropriate antibiotic, and incubated for 5 days at 30°C under microaerobic conditions.

## A. butzleri RM4018 ΔAbu0127

The *A. butzleri Abu0127* deletion construct (pUC18/ΔAbu0127) was prepared as follows (Fig. S12A) (69). The upstream and downstream regions of *Abu0127* were amplified by PCR using the P10-P11 and P14-P15 primer pairs, respectively, and an *A. butzleri* RM4018 genomic DNA as a template. The *aphA-3* cassette was amplified using the P12-P13 primer pair; pTZ57R/TΔHP1021 (23) was used as a template. The resulting fragments were purified on an agarose gel. Subsequently, the PCR-amplified fragments and the SmaI digested vector pUC18 were ligated according to the method of Gibson et al. (67). Subsequently, *A. butzleri* RM4018 was transformed via electroporation with the pUC18/ΔAbu0127 plasmid, and the transformants were selected by plating on CBA-B plates supplemented with kanamycin. The allelic exchange was verified by PCR using the P16-P17 primer pair. The lack of *Abu0127* in *A. butzleri* RM4018 ΔAbu0127 was confirmed by RNA-seq (Fig. S12B), LC-MS/MS (S2 Data), and Western blot (Fig. S12C and D).

## Disk diffusion assay

Bacteria were cultured in BHI to $OD_{600}$ = ~0.5 to 0.7 and then diluted to $OD_{600}$ = 0.1 in BHI. Each culture was spread by a cotton swab on CBA-B plates. Sterile, glass fiber 6-mm disks were placed on plates, and 5 µL of tested solutions were dropped on disks: 2% $H_2O_2$ (POCH, 885193111), 2% paraquat dichloride (Acros Organics, 227320010), 5% hydroxyurea (Merck, H8627-1G), 3% sodium nitroprusside (Merck, S0501), 2.5% menadion (Merck, A13593), and 15% sodium hypochlorite (Merck, 1056142500). Sodium hypochlorite solutions for assays were made fresh; the solutions' pH was adjusted by adding HCl to pH 7 and kept in phosphate-buffered saline (PBS) buffer prior to experiments (70). Menadion solution was prepared in dimethyl sulfoxide (DMSO). The diameter of the inhibition zone around the disks was determined after 3 days of incubation under microaerobic conditions. The experiment was performed using three biological replicates.

## RNA isolation

Bacterial cultures (12-mL BHI) of *C. jejuni* and *A. butzleri* strains were grown under microaerobic conditions to $OD_{600}$ of 0.5–0.6. Immediately after opening the jar, 2 mL of the non-stressed culture was added to 2 mL of the RNAprotect Bacteria Reagent (Qiagen), vortexed, and incubated for 5 min at room temperature. In parallel, the cultures were treated with 1-mM paraquat (final concentration) for 25 min (*C. jejuni* 42℃, *A. butzleri* 30℃, 140-rpm orbital shaking). After oxidative stress, samples were collected similarly to non-stressed cells. After 5-min incubation with RNAprotect Bacteria Reagent, bacteria were collected by centrifugation (4,700 × *g*, 10 min, room temperature). RNA was isolated by GeneJET RNA Purification Kit (Thermo Fisher Scientific, K0731) according to the manufacturer's protocol and treated with RNase-free DNase I (Thermo Fisher Scientific). Next, purification by the GeneJET RNA Purification Kit was performed to remove DNase I. A NanoDrop Lite spectrophotometer, agarose gel electrophoresis, and Agilent 4200 TapeStation System were used to determine the RNA quality and quantity. RNA was isolated immediately after bacteria collection, stored at −80℃ for up to 1 month and used for RNA sequencing. RNA was isolated from three independent cultures.

For analyses of gene transcription dependent on oxygen supply, bacterial cells were collected from cultures under the logarithmic phase of growth ($OD_{600}$ ~0.2 to 0.5), grown under the desired atmosphere (see Materials and Culture Conditions), and RNA was isolated as described above.

## RNA-seq

Preparation and sequencing of the prokaryotic directional mRNA library were performed at the Novogene Bioinformatics Technology Co. Ltd. (Cambridge, UK). Briefly, the ribosomal RNA was removed from the total RNA, followed by ethanol precipitation. After fragmentation, the first strand of cDNA was synthesized using random hexamer primers. During the second-strand cDNA synthesis, deoxyuridine triphosphates were replaced with deoxythymidine triphosphates in the reaction buffer. The directional library was ready after end repair, A-tailing, adapter ligation, size selection, USER enzyme digestion, amplification, and purification. The library was checked with Qubit, RT-qPCR for quantification, and a bioanalyzer for size distribution detection. The libraries were sequenced with the NovaSeq 6000 (Illumina), and 150-bp reads were produced.

## RNA-seq analysis

The 150-bp paired reads were mapped to the *C. jejuni* NCTC 11168 (NC_002163.1) or *A. butzleri* RM4018 (NC_009850.1) genome depending on the species analyzed using Bowtie2 software with local setting (version 2.3.5.1) (71, 72) and processed using samtools (version 1.10) (73), achieving more than $10^6$ mapped reads on average. Differential analysis was performed using R packages Rsubread (version 2.10) and edgeR (version 3.38) (74, 75), following a protocol described in reference (76). Genes rarely

transcribed were removed from the analysis (less than 10 mapped reads per library). The obtained count data were normalized using the edgeR package, and a quasi-likelihood negative binomial was fitted. Differential expression was tested using the glmTtreat function with a 1.45-fold change (FC) threshold. Only genes with a false discovery rate (FDR) less than 0.05 and $|\log_2 FC|$ of $\geq 1$ were considered differentially expressed. Data visualization with volcano plots was done using the EnhancedVolcano and tidyHeatmap R packages (versions 1.14 and 1.8.1) (77). The reproducibility of *C. jejuni* and *A. butzleri* biological replicates was visualized by principal component analysis (PCA) of the normalized RNA-seq CPM data (Fig. S13A and B).

## Proteomic sample preparation

Bacterial cultures of wild-type *C. jejuni* NCTC 11168 and *A. butzleri* RM4018 strains and their isogenic mutant strains, ΔCj1608 or ΔAbu0127, respectively, were grown in BHI under microaerobic conditions to an $OD_{600}$ of 0.5–0.7 and split into two subcultures of 10 mL each. The first subculture was harvested, washed, and lysed immediately after opening the jar; the second subculture (only WT strains) was harvested, washed, and lysed after incubation with 1-mM paraquat (*C. jejuni*: 42°C, *A. butzleri*: 30°C, with orbital shaking at 140 rpm) for 30 and 60 min. The bacterial proteomes were prepared as described previously by Abele et al. (78). Briefly, cells were harvested by centrifugation at $10,000 \times g$ for 2 min; media were removed; and cells were washed once with 20 mL of 1 × PBS. The cell pellets were suspended and lysed in 100 µL of 100% trifluoroacetic acid (Roth) (79) for 5 min at 55°C. Next, 900 µL of neutralization buffer (2-M Tris) was added and vortexed. Protein concentration was measured using Bradford assay (Bio-Rad). Fifty micrograms of protein per sample was reduced [9-mM tris(2-carboxyethyl)phosphine] and carbamidomethylated (40 mM chloroacetamide) for 5 min at 95°C. The proteins were digested by adding trypsin (proteomics grade, Roche) at a 1:50 enzyme:protein ratio (wt/wt) and incubation at 37°C overnight. Digests were acidified by the addition of 3% (vol/vol) formic acid (FA) and desalted using self-packed StageTips (five disks per microcolumn, ø 1.5 mm, C18 material; 3M Empore). The peptide eluates were dried to completeness and stored at −80°C. Before the LC-MS/MS measurement, all samples were freshly resuspended in 12-µL 0.1% FA in high-performance liquid chromatography (HPLC)-grade water, and around 25 µg of total peptide amount was injected into the mass spectrometer per measurement. Each experiment was performed using four biological replicates.

## Proteomic data acquisition and data analysis

Peptides were analyzed on a Vanquish Neo liquid chromatography system (microflow configuration) coupled to an Orbitrap Exploris 480 mass spectrometer (Thermo Fisher Scientific). Around 25 µg of peptides was applied onto an Acclaim PepMap 100 C18 column (2-µm particle size, 1-mm ID × 150 mm, 100-Å pore size; Thermo Fisher Scientific) and separated using a two-step gradient. In the first step, a 50-min linear gradient ranging from 3% to 24% solvent B (0.1% FA, 3% DMSO in acetonitrile) in solvent A (0.1% FA and 3% DMSO in HPLC-grade water) at a flow rate of 50 µL/min was applied. In the second step, solvent B was further increased from 24% to 31% over a 10-min linear gradient. The mass spectrometer was operated in data-dependent acquisition and positive ionization modes. MS1 full scans (360–1,300 *m/z*) were acquired with a resolution of 60,000, a normalized automatic gain control (AGC) target value of 100%, and a maximum injection time of 50 ms. Peptide precursor selection for fragmentation was carried out using a fixed cycle time of 1.2 s. Only precursors with charge states from 2 to 6 were selected, and dynamic exclusion of 30 s was enabled. Peptide fragmentation was performed using higher-energy collision-induced dissociation and a normalized collision energy of 28%. The precursor isolation window width of the quadrupole was set to 1.1 *m/z*. MS2 spectra were acquired with a resolution of 15,000, a fixed first mass of 100 *m/z*, a normalized AGC target value of 100%, and a maximum injection time of 40 ms.

Peptide identification and quantification were performed using MaxQuant (version 1.6.3.4) with its built-in search engine Andromeda (80, 81). MS2 spectra were searched against the *C. jejuni* or *A. butzleri* proteome database derived from *C. jejuni* NCTC 11168 (NC_002163.1) or *A. butzleri* RM4018 (NC_009850.1), respectively. Trypsin/P was specified as the proteolytic enzyme. The precursor tolerance was set to 4.5 ppm, and fragment ion tolerance was set to 20 ppm. Results were adjusted to a 1% FDR on peptide spectrum match level and protein level employing a target-decoy approach using reversed protein sequences. The minimal peptide length was defined as seven amino acids; carbamidomethylated cysteine was set as fixed modification and oxidation of methionine and N-terminal protein acetylation as variable modifications. The match-between-run function was disabled. Protein abundances were calculated using the label-free quantification (LFQ) algorithm from MaxQuant (82). Protein intensity values were logarithm transformed (base 2), and a Student *t*-test was used to identify proteins differentially expressed between conditions. The resulting *P* values were adjusted by the Benjamini-Hochberg algorithm (83) to control the FDR. Since low abundant proteins are more likely to result in missing values, we filled in missing values with a constant of half the lowest detected LFQ intensity per protein. However, if the imputed value was higher than the 20% quantile of all LFQ intensities in that sample, we used the 20% quantile as the imputed value. Only proteins with $|\log_2FC|$ of $\geq 1$ were considered differentially expressed (84).

The reproducibility of *C. jejuni* and *A. butzleri* biological replicates was visualized by PCA of the normalized LC-MS/MS CPM data (Fig. S13C and D).

## Omics data analysis

The KEGG (85) gene/protein set enrichment analysis of the differentially expressed genes and proteins was performed and visualized based on the clusterProfiler (29) (version 4.8.2) package in R software with a *P* value of < 0.05. Genome-wide functional annotation was carried out with the eggNOG-mapper (version 5.0) (86) according to the COG database (87) with an *e* value of 0.001 and a minimum hit bit score of 60. Visual representations of RNA expression levels and LC-MS/MS expressed proteins in different COGs were performed with the Circos Table viewer (http://mkweb.bcgsc.ca/tableviewer/, accessed on 5 November 2023). For the analyses, only genes and proteins with a $|\log_2FC|$ of $\geq 1$ and an FDR of < 0.05 were considered as significantly changed.

## ChIP

Bacterial cultures (70- mL BHI) of *C. jejuni* NTCT 11168 and *A. butzleri* RM4018 wild-types and ΔCj1608 and Abu0127 mutants were grown to an $OD_{600}$ of 0.5–0.7. The culture was cross-linked with 1% formaldehyde for 5 min immediately after opening the jar. The cross-linking reactions were stopped by treatment with 125 mM glycine for 10 min at room temperature. The cultures were centrifuged at 4,700 × *g* for 10 min at 4°C and washed twice with 25 mL of ice-cold 1 × PBS, followed by the same centrifugation step. Samples were resuspended in 1.1- mL immunoprecipitation (IP) buffer (150-mM NaCl, 50-mM Tris-HCl, pH 7.5, 5-mM EDTA, 0.5% vol/vol NP-40, and 1.0% vol/vol Triton X-100) and sonicated [Ultraschallprozessor UP200s (0.6%/50% power, 30-s on, 0-s off, ice bucket)] to reach a 100- to 500-bp DNA fragment size. Next, the samples were centrifuged at 12,000 × *g* for 10 min at 4°C. One hundred microliters of the supernatant was used for input preparation. Nine hundred microliters of the supernatant was incubated with 30 µL of Sepharose Protein A (Rockland, PA50-00-0002) (pre-equilibrated in IP buffer) for 1 h at 4°C on a rotation wheel. The samples were centrifuged at 1,000 × *g* for 2 min at 4°C. The supernatants were incubated with 100-µL antibody-Sepharose A complex (see below) and incubated at 4°C for 24 h on a rotation wheel. Next, the samples were centrifuged at 1,000 × *g* for 2 min at 4°C, and the supernatant was discarded. The beads were washed four times with IP-wash buffer (50-mM Tris-HCl, pH 7.5, 150-mM NaCl, 0.5% NP-40, 0.1% SDS), twice with Tris-EDTA (TE) buffer (10-mM Tris-HCl, pH 8.0; 0.1-mM EDTA), resuspended in 180 µL of TE buffer, and treated with 20-µg/mL RNase A at

37°C for 30 min. Next, cross-links were reversed by adding sodium dodecyl sulfate (SDS) at a final concentration of 0.5% and proteinase K at a final concentration of 20 µg/mL, followed by incubation for 16 h at 37°C. The beads were removed by centrifugation at 1,000 × $g$ for 2 min at 4°C, and the DNAs from the supernatants were isolated with ChIP DNA Clean & Concentrator (Zymo Research). The quality of DNA was validated by electrophoresis in 2% agarose gel, and the concentration was determined with QuantiFluor dsDNA System (Promega). The ChIP-DNA was isolated from three independent bacteria cultures.

The *C. jejuni* antibody-Sepharose A complex was prepared by adding 40 µg of desalted rabbit polyclonal anti-Cj1608 antibody to 100 µL of the Sepharose Protein A pre-equilibrated in IP buffer. *A. butzleri* antibody-sepharose A complex was prepared by adding 120 µg of desalted rabbit polyclonal anti-Abu0127 antibody to 100 µL of the Sepharose Protein A pre-equilibrated in IP buffer. The binding reaction was performed on a rotation wheel for 24 h at 4°C. Next, the complex was washed five times with an IP buffer. The antibodies used in ChIP were raised in rabbits and validated with Western blot (Fig. S11D and D).

## Quantitative polymerase chain reaction

RT-qPCR quantified the mRNA levels of the selected genes. The reverse transcription was conducted using 500 ng of RNA in a 20-µL volume reaction mixture of iScript cDNA Synthesis Kit (Bio-Rad). Diluted cDNA (1:10, 2.5 µL) was added to 7.5 µL of Sensi-FAST SYBR No-ROX (Bioline) and 400 nM of forward and reverse primers in a 15-µL final volume. The RT-qPCR program was 95°C for 3 min, followed by 40 three-step amplification cycles consisting of 5 s at 95°C, 10 s at 58°C, and 20 s at 72°C. The following primer pairs were used: P18-P19, *gltA* for *C. jejuni* RT-qPCR, and P23-P24, *nuoB* for *A. butzleri* RT-qPCR (Table S2). The relative quantity of mRNA for each gene of *C. jejuni* was determined by referring to the mRNA levels of *recA* (P25-P26 primer pair). The relative quantity of mRNA for each gene of *A. butzleri* was determined by referring to the mRNA levels of *gyrA* (P27-P28 primer pair). The RT-qPCR was performed for three independent bacterial cultures.

The protein-DNA interactions in the cell *in vivo* of the selected DNA regions were quantified by ChIP-qPCR. Diluted immunoprecipitation output (1:0, 2.5 µL) was added to 7.5 µL of Sensi-FAST SYBR No-ROX (Bioline) and 400 nM of forward and reverse primers in a 15-µL final volume. The ChIP-qPCR was performed using the following program: 95°C for 3 min, followed by 40 three-step amplification cycles consisting of 10 s at 95°C, 10 s at 59°C, and 20 s at 72°C. The following primer pairs were used: P43-P44, p*gltA* (*C. jejuni*), and P45-P46, p*nuoA* (*A. butzleri*) (Table S2). The *recA* (P25-P26) and *gyrA* (P27-P28) genes were used as a negative control for *C. jejuni* and *A. butzleri*, respectively (Table S2). No-antibody control was used for ChIP-qPCR normalization, and the fold enrichment was calculated. The ChIP-qPCR was performed for three independent bacterial cultures.

The RT-qPCR and ChIP-qPCR were performed using the CFX96 Touch Real-Time PCR Detection Systems, and data were analyzed with CFX Maestro (Bio-Rad) software.

## Construction of plasmids expressing recombinant wild-type Cj1608 and Abu0127 proteins

The *Cj1608* (*Cj1509* in *C. jejuni* 81–116) gene (888 bp) was amplified with primer pair P29-P30 using *C. jejuni* 81–116 genomic DNA as a template. The *Abu0127* gene (891 bp) was amplified with primer pair P31-P32 using *A. butzleri* RM4018 genomic DNA as a template. The PCR products were digested with BamHI/SalI and cloned into BamHI/SalI sites of pET28Strep (24) to generate pETStrepCj1509 and pETStrepAbu0127, respectively (Table S1).

## Protein expression and purification

The recombinant Strep-tagged Cj1608 (Cj1509 of *C. jejuni* 81–116) and Abu0127 proteins were purified according to the Strep-Tactin manufacturer's protocol (IBA Lifesciences).

Briefly, *E. coli* BL21 cells (1 L) carrying either the pET28/StrepCj1509 or pET28/Stre-pAbu0127 vectors were grown at 37°C. At an $OD_{600}$ of 0.8, protein synthesis was induced with 0.05-mM IPTG for 3 h at 30°C for Cj1608 and overnight at 18°C for Abu0127. The cultures were harvested by centrifugation (10 min; 5,000 × *g*; at 4°C). The cells were suspended in 20 mL of ice-cold buffer W (100-mM Tris-HCl, pH 8.0; 300-mM NaCl; and 1-mM EDTA) supplemented with cOmplete, EDTA-free protease inhibitor Cocktail (Roche), disrupted by sonication, and centrifuged (30 min; 31,000 × *g*; at 4°C). The supernatant was applied onto a Strep-Tactin Superflow high-capacity Sepharose column (1-mL bed volume, IBA). The column was washed with buffer W until Bradford tests yielded a negative result, and then washed with 5 mL of buffer W without EDTA. The elution was carried out with approximately 6 × 0.8 mL of buffer E (100-mM Tris-HCl, pH 8.0; 300-mM NaCl; and 5-mM desthiobiotin). Protein purity was analyzed by SDS-PAGE electrophoresis using GelDoc XR+ and ImageLab software (Bio-Rad). The fractions were stored at −20°C in buffer E diluted with glycerol to a final concentration of 50%.

## EMSA

PCR amplified DNA probes in two steps. DNA fragments were amplified in the first step using unlabeled primer pairs P33-P34 (p*gltA*) and P35-P36 (*recJ*) with a *C. jejuni* NCTC 11168 genomic DNA template, and P37-P38 (p*nuo*) and P39-P40 (*recA*) with an *A. butzleri* RM4018 genomic DNA template (Table S1). The forward primers were designed with overhangs complementary to P41 (FAM labeled) for p*nuo* and p*gltA* and P42 (Cy5 labeled) for *recJ* and *recA*. The unlabeled fragment was purified and used as a template in the second round of PCR using the appropriate fluorophore-labeled primer and the appropriate reverse primer used in the first step. Both fluorophore-labeled DNAs (each 5 nM) were incubated with the Strep-tagged protein at 30°C (Abu0127) or 37°C (Cj1608) for 20 min in Tris buffer (50-mM Tris-HCl, pH 8.0; 100-mM NaCl; and 0.2% Triton X-100). The complexes were separated by electrophoresis on a 4% polyacrylamide gel in 0.5 × Tris-Borate-EDTA (TBE) (1× TBE: 89-mM Tris, 89-mM borate, and 2-mM EDTA) at 10 V/cm in the cold room (approximately 10°C). The gels were analyzed using Typhoon 9500 FLA Imager and ImageQuant software.

## ATP assay

The ATP level was measured with the BacTiter-Glo Assay. *C. jejuni and A. butzleri* cultures were grown in the BHI medium to the logarithmic growth phase ($OD_{600}$ of 0.5–0.7). Cells were diluted to an $OD_{600} = 0.1$ with fresh medium. Next, 50 µL of bacteria was mixed with 50 µL of BacTiter-Glo (Promega) and incubated at room temperature for 5 min. The luminescence was measured with a CLARIOstar plate reader on opaque-walled multi-well plates (SPL Life Sciences, 2–200203). Each experiment was performed using three biological replicates.

## Statistics and reproducibility

Statistical analysis was performed using GraphPad Prism (version 8.4.2) and R (version 4.3.1) statistical software. All *in vivo* experiments were repeated at least three times, and data were presented as mean ± SD. The statistical significance between the two conditions was calculated by paired two-tailed Student *t*-test. The statistical significance between multiple groups was calculated by one-way analysis of variance (ANOVA) with Tukey's post hoc test. A *P* value of < 0.05 was considered statistically significant. The proteome and transcriptome correlations were determined with the Pearson correlation coefficient. The EMSA and Western blot experiments were repeated twice with similar results.

## ACKNOWLEDGMENTS

We thank Kinga Surmacz for helping with disk diffusion assays, Maria Cieślak for helping with pUC18/ΔAbu0127 plasmid preparation, and Verena Breitner and Tayma Midari for technical assistance at BayBioMS.

This work has been supported by the OPUS 17 (project number 2019/33/B/NZ6/01648, funded by the National Science Centre, Poland, to A.Z.-P.) and by the EPIC-XS (project number 823839, funded by the Horizon 2020 Program of the European Union, to C.L.).

The open-access publication of this article was funded by the OPUS 17 (2019/33/B/NZ6/01648, National Science Centre, Poland) and statutory funds from the Ludwik Hirszfeld Institute of Immunology and Experimental Therapy, Polish Academy of Sciences.

## AUTHOR AFFILIATIONS

[1]Department of Microbiology, Hirszfeld Institute of Immunology and Experimental Therapy, Polish Academy of Sciences, Wrocław, Poland

[2]Department of Molecular Microbiology, Faculty of Biotechnology, University of Wrocław, Wrocław, Poland

[3]Department of Biological Safety, Unit of Product Hygiene and Disinfection Strategies, German Federal Institute for Risk Assessment, Berlin, Germany

[4]Bavarian Center for Biomolecular Mass Spectrometry (BayBioMS), Technical University of Munich (TUM), Freising, Germany

[5]Department of Biological Safety, National Reference Laboratory for *Campylobacter*, German Federal Institute for Risk Assessment, Berlin, Germany

## AUTHOR ORCIDs

Mateusz Noszka  http://orcid.org/0000-0002-6694-1394
Agnieszka Strzałka  http://orcid.org/0000-0002-7092-0609
Jakub Muraszko  http://orcid.org/0000-0001-7232-0298
Dirk Hofreuter  http://orcid.org/0000-0002-2589-9206
Miriam Abele  http://orcid.org/0000-0003-0084-2999
Christina Ludwig  http://orcid.org/0000-0002-6131-7322
Kerstin Stingl  http://orcid.org/0000-0002-8338-717X
Anna Zawilak-Pawlik  http://orcid.org/0000-0003-1824-1550

## FUNDING

| Funder | Grant(s) | Author(s) |
| --- | --- | --- |
| Narodowe Centrum Nauki (NCN, National Science Centre, Poland) | OPUS 17 2019/33/B/NZ6/01648 | Anna Zawilak-Pawlik |
| EC \| Horizon 2020 Framework Programme (H2020) | EPIC-XS Project Number 823839 | Christina Ludwig |

## AUTHOR CONTRIBUTIONS

Mateusz Noszka, Conceptualization, Data curation, Formal analysis, Funding acquisition, Investigation, Methodology, Validation, Visualization, Writing – original draft, Writing – review and editing, Software, Supervision | Agnieszka Strzałka, Formal analysis, Methodology, Software, Writing – review and editing, Supervision, Investigation, Visualization, Validation | Jakub Muraszko, Investigation, Writing – review and editing | Dirk Hofreuter, Validation, Writing – review and editing, Formal analysis | Miriam Abele, Data curation, Methodology, Writing – review and editing, Validation | Christina Ludwig, Funding acquisition, Methodology, Writing – review and editing | Kerstin Stingl, Conceptualization, Methodology, Writing – review and editing, Formal

analysis, Project administration, Validation | Anna Zawilak-Pawlik, Conceptualization, Formal analysis, Funding acquisition, Project administration, Supervision, Validation, Visualization, Writing – original draft, Writing – review and editing, Data curation

## DATA AVAILABILITY

The *Campylobacter jejuni* RNA-seq FASTQ and processed data generated in this study have been deposited in the ArrayExpress database (EMBL-EBI) under accession code E-MTAB-13650. The *Arcobacter butzleri* RNA-seq FASTQ and processed data generated in this study have been deposited in the ArrayExpress database (EMBL-EBI) under accession code E-MTAB-13649. The raw proteomics data, MaxQuant search results, and the used protein sequence database generated in this study have been deposited in the ProteomeXchange Consortium via the PRIDE partner repository (88) under accession code PXD048711. *Campylobacter jejuni* NCTC 11168 reference genome is deposited in the National Center for Biotechnology Information under accession code NC_002163.1. *Arcobacter butzleri* RM4018 reference genome is deposited in the National Center for Biotechnology Information under accession code NC_009850.1. All of the code and data used to generate the figure presented here are deposited in GitHub via https://github.com/NoszkaM/LBMM.git.

## ETHICS APPROVAL

The antibodies used in chromatin immunoprecipitation and western blot were raised in rabbits under the approval of the First Local Committee for Experiments with the Use of Laboratory Animals, Wroclaw, Poland (consent number 053/2020/P2).

## ADDITIONAL FILES

The following material is available online.

### Supplemental Material

**Data S1 (mSystems00784-24-s0001.xlsx).** Full list of genes and proteins of the Cj1608 regulon.
**Data S2 (mSystems00784-24-s0002.xlsx).** Full list of genes and proteins of the Abu0127 regulon.
**Data S3 (mSystems00784-24-s0003.xlsx).** Genes and proteins of selected processes or pathways in *C. jejuni*, *A. butzleri*, and *H. pylori*.
**Supplemental material (mSystems00784-24-s0004.pdf).** Figures S1 to S13, Tables S1 and S2, and descriptions of Data S1 to S3.

### Open Peer Review

**PEER REVIEW HISTORY (review-history.pdf).** An accounting of the reviewer comments and feedback.

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
