## [Reviewer comments · mSystems]

CemR atypical response regulator impacts energy conversion in *Campylobacter*

Mateusz Noszka, Agnieszka Strzałka, Jakub Muraszko, Dirk Hofreuter, Miriam Abele, Christina Ludwig, Kerstin Stingl, and Anna Pawlik

Corresponding Author(s): Anna Pawlik, Hirsfeld Institute of Immunology and Experimental Therapy PAS

Review Timeline:

Submission Date:	June 9, 2024
Editorial Decision:	June 10, 2024
Revision Received:	June 11, 2024
Accepted:	June 12, 2024

Editor: Jack Gilbert

Reviewer(s): The reviewers have opted to remain anonymous.

Transaction Report:

DOI: <https://doi.org/10.1128/msystems.00784-24>

Re: mSystems00784-24 (CemR atypical response regulator impacts energy conversion in *Campylobacter*)

Dear Dr. Anna Magdalena Pawlik:

Revision Guidelines

Sincerely,
Jack Gilbert
Editor
mSystems

Re: mSystems00784-24R1 (CemR atypical response regulator impacts energy conversion in *Campylobacter*)

Dear Dr. Anna Magdalena Pawlik:

Your manuscript has been accepted, and I am forwarding it to the ASM production staff for publication. Your paper will first be checked to make sure all elements meet the technical requirements. ASM staff will contact you if anything needs to be revised before copyediting and production can begin. Otherwise, you will be notified when your proofs are ready to be viewed.

Sincerely,

Jack Gilbert
Editor
mSystems